# Factorial Design with Simulation for the Optimization of the Level of Service in the Platform-Train Interface of Metro Stations—A Pilot Study

**Matias Kulczewski [1], Andres Wilson [1], Sebastian Seriani [2,\*] and Taku Fujiyama [3]**

[1]  Facultad de Ingeniería y Ciencias Aplicadas, Universidad de los Andes, Santiago de Chile 7620001, Chile
[2]  Escuela de Ingeniería de Construcción y Transporte, Pontifica Universidad Católica de Valparaíso, Valparaíso 2362804, Chile
[3]  Faculty of Civil, Environmental and Geomatic Engineering, University College London, Chadwick Building, Gower St., London WC1E 6BT, UK
\*  Correspondence: sebastian.seriani@pucv.cl

**Abstract:** Metro stations are considered complex areas of pedestrian mobility due to the increasing congestion, due to the a high level of demand of different circulation spaces. Given this situation and the limited physical spaces remaining to develop transport systems in urban areas, railways acquire greater relevance given the need to mobilize pedestrians. Within the stations, the most problematic area is the platform-train interface (PTI) due to the high number of interactions between passengers boarding and alighting. The objective of this study is to identify the PTI configuration that maximizes the level of service for passengers, safeguards the operational continuity of the system and improves user experience by reducing dissatisfaction and delay times. For this, a pedestrian microsimulation model is used in order to recreate the reality of a generic metro station and its different scenarios given the combinations of two factors: the platform configurations (topology) and the traffic control elements. Subsequently, these scenarios are analyzed through a factorial design, looking for the situation that optimizes the combination of metrics chosen in a weighted way. Finally, it is found that the PTI configuration that maximizes the level of service for users is the mixed station with signaling. It is this which includes the factors with the greatest positive effect on the chosen metrics.

**Keywords:** passenger; metro station; level of service; simulation; platform-train interface; optimization

## 1. Introduction

Metro stations are considered complex areas of pedestrian mobility due to the existence of different circulation spaces that make up the route for pedestrians to access the metro service. As a result of the different flow routes within these spaces, conflict areas are created that congest the train system during peak times with the greatest demand. Like other public transport systems, the metro follows a circuit made up of restrictions, which correspond to the stations. The stations restrict the speed of the system, which directly affects its transport capacity, which serves as a measure of the efficiency of a station [1].

Currently, there are simulation tools that allow metro stations to be modeled for later analysis. The pedestrian microsimulation is very useful approach for studying the behaviors of individuals (or crowds) in different scenarios, using variables such as pedestrian flow, waiting time on platforms, location, specific behavior trends, etc. This seeks to observe how the changes in the variables affect the simulation result to compare the different scenarios and finally obtain the relevant information to know how to change these spaces in order to optimize the designs [2,3].

According to Seriani and Fernandez [4], metro stations can be divided into five places: the platform-train interface (PTI), platform-stairs, mezzanine, complementary (for example, commerce) and city. The authors used a microsimulation tool to study the different spaces,

in which the PTI is considered the area where the greatest number of interactions between passengers take place, during the boarding and alighting process. In the PTI, there is a large volume of interactions and counterflows. During peak times of demand, this area tends to collapse in the stations with a high flow of passengers, which increases the risk of accidents and generates dissatisfaction due to the delay and inconvenience passengers suffer.

The problem arises since, in Chile and Latin America, the station system has been created with a more architectural logic than an engineering approach, which generates inadequate designs in operational terms. This is reflected in how inefficient stations are with respect to the level of service in the PTI [1]. In the subway stations of Santiago de Chile, usually during peak hours, the maximum pedestrian flow capacity is exceeded, which corresponds to the critical level of service. Therefore, there is congestion in the PTI, which generates discomfort, dissatisfaction, and an increase in the perception of risk.

Similarly, in the case of the United Kingdom, more than three billion interactions on the train network are reached each year, in which 48% of the risk of fatalities for passengers occur in the PTI [5,6]. Therefore, this complex space presents different risks and dangers for passengers. Accidents can occur during boarding and alighting or simply at the edge of the platform when passengers wait for the train to arrive.

Given this problem, some measures have been taken to manage flows in the Santiago Metro [7], where boarding was implemented on both platforms for the Tobalaba station (Line 4) during peak hours. In addition, other projects have been implemented, such as a central platform at the Vicuña Mackenna station (Line 4), a programmed entry control at stations (ticket office, platforms and accesses), the installation of a yellow safety line on the edge of the tracks and the extension of the network with new and automatized metro lines, among others. However, these measures have not been enough to fully mitigate the increase in flow at peak hours and improve the level of service at the PTI [8,9]. The fact that only side platforms are used in metro stations in Santiago de Chile makes it logical (even necessary) to analyze other types of configurations and study possible measures for flow management, since the effect that these can have on passenger behavior is currently unknown, being the main objective of this study.

Therefore, the purpose of this study is to analyze more suitable design alternatives for new metro stations and propose modifications that allow for increasing the operational performance of current facilities by comparing and identifying the PTI configurations and design through a pedestrian microsimulation tool that favors the rapid and safe boarding and alighting of pedestrians. Thus, there is both a theoretical and a social motive in this research. The first reveals new representative statistical data, in order to add value to the fields of study on pedestrian behavior and to present new ways of managing pedestrian spaces in metro stations. The second, social motive is to identify the PTI configuration that maximizes the level of service for passengers, that safeguards the operational continuity of the system and, thereby, improves the travel experience for users by reducing dissatisfaction and delay time in the system.

The structure of the paper is divided into 6 sections. In Section 2, different studies of passenger behavior are reported. Next, in Section 3, the simulation method is defined. In Section 4, the results are analyzed, followed by the discussion in Section 5 and the conclusions in Section 6.

## 2. Passenger Behavior in the PTI

According to RSSB [5], passenger behavior can be affected by four factors in public transport systems such as metro stations: the presence of other people (density on the platform or personal space), the physical design of the car of the train (width of the platform, number of doors of the train or position of the seats, etc.), the information provided to pedestrians (maps, on-board screens, markings on the floor, etc.) and the environment (weather).

Other factors studied by Still [2] can also be considered, such as gender, size, age and other characteristics that influence how individuals behave, with which different types of

pedestrian profiles are generated. Similarly, there are psychological profiles that define the behavior of each passenger and his/her preferences (e.g., cultural factors).

Of these factors, those that can alter the behavior in the PTI and that are controllable by means of simulation for the analysis of different configurations are considered, in which interaction is understood as the way in which the behavior of pedestrians is related to the physical environment and to other passengers [1]. Therefore, the presence of other passengers, the physical layout of the train cars and the implementation of information that provides directions are all considered in the current study.

There are several authors who analyze pedestrians and their behavior, but many consider Fruin [10] to be one of the most recognized in the investigation of crowd behavior by introducing the level of service (LOS). Reviewing earlier studies, Still [2,3] paraphrases Fruin by considering body depth and shoulder width as the the primary human measurements used when considering pedestrian spaces and facilities. The width of the shoulders is the main factor in the design of doors and stairs. Many doorways are designed to allow two or more people to come forward, but are not really wide enough for this purpose. From these studies, it was concluded that on average a working man has a length of 53 cm in shoulder width and 28 cm in body width. Another way in which standardizations define the profiles is the sensory zone that each pedestrian has, which comprises an area that reaches approximately 1.48 m and an average walking speed of 1.37 m/s [10]. It is important to understand that each individual, according to their respective characteristics, has different levels of satisfaction according to the existing pedestrian density (passengers/m$^2$) and the environment through which they circulate. For example, people with disabilities require a lower pedestrian density and a greater separation space from other pedestrians around them to feel comfortable [11].

The LOS is introduced by Fruin [10] to address the capacity of the design or the provision of space in public transport spaces. The LOS is used to study the level of pedestrian flow that occurs in certain scenarios; that is, in areas where a certain domain of design is allowed, such as in urban trains, in which high levels of LOS can be provided with the consequent improvement of the pedestrian environment. It should also be mentioned that LOS corresponds to a quantitative indicator of the performance measurement that represents the quality of service, such as travel time, speed, delay, comfort, convenience, safety, and cost, among others. The set of measurements used to determine the LOS in the elements of a transport system is called a service measure. The LOS is used as a method to indicate the degrees of congestion and conflicts in study areas such as flat areas, queues (waiting areas) or stairs, through general parameters such as speed, density or flow. Fruin [10] studied the behavior of passengers in some metro stations to obtain the capacity of a journey. With this information, he reported that, by reducing the space, the flow increases up to the limited capacity of the space and, in turn, the movement of passengers is reduced. That is, the higher the density, the lower the flow and speed of passengers. The Highway Capacity Manual (HCM) [12] defines six LOS, ranging from letter A to letter F, for each service measure. Level A represents the best-operating conditions of the space from the passengers' point of view (free flow without conflicts); then, the other levels represent a greater amount of flow; that is, a greater density per m$^2$, which generates a decrease in traffic speed; Level E is equivalent to the total capacity of the space considered in the design. Finally, Level F is defined as the critical density in which the design occupancy is exceeded (Highway Capacity Manual, 2010). Fruin [10] used the LOS method and was able to study the density of people, the space they used, the speed at which passengers moved and the level of occupancy they used in the public transport vehicle. He also showed that the LOS can be divided for passengers who are moving or in waiting areas (for example, inside at the PTI), according to the density, space and percentage of occupancy in the vehicle.

In conclusion, the LOS is an effective methodology to study the different densities of people, but it is not for the analysis of the distribution of passengers in such crowded spaces as the PTI. Although the LOS measures the density in the train and on the platform, it cannot be used to measure the distribution that the passengers will have in the different

zones inside the train. Some authors have shown the need to include other parameters complementing the LOS. According to Evans and Wener [13], the overall density used in the LOS does not predict which space presents more interaction between passengers. The authors studied density, stress and commuting in trains where passengers must be seated next to others and found that the level of stress increased as the density went up. Kaparias et al. [14] studied the experience of pedestrians. The authors reported that existing studies have highlighted relevant factors that specifically affect the walking experience, such as Sarkar's [15] level of service, which is based on safety, comfort and convenience, continuity, consistency system and attractiveness, or the study by Pikora et al. [16] in which the quality of walking depends on functional, safety, aesthetic and destination factors. The authors [14] evaluated the environment and the factors that specifically affect the experience of pedestrians based on questionnaires and regression models, following the PERS software [17].

The LOS has been studied in metro stations, highlighting its application to the PTI, which corresponds to the area where passengers circulate to get on and off the train, one of the most complex movements within metro stations [18]. It is so complex due to the large number of interactions between pedestrians generated in this space, which can result in congestion problems, operational inefficiency and accidents, which, in the worst case, can be fatal. This place is usually made up of the following physical design components: the train, the train doors, the vertical/horizontal gap between the train and the platform, the yellow line on the platform, the platform waiting areas, the seats and the platform edge doors, among others.

Within this space, Seriani and Fernandez [4] identify three categories of variables that model the behavior and interaction between passengers: physical, spatial, and operational. Physical variables are defined as those that are specifically related to the dimensions of physical components (e.g., platform width). Spatial variables are considered as the elements of circulation that can be used to change the behavior of passengers (e.g., platform edge doors). The operational variables are focused on the uncontrollable characteristics of passenger movement (e.g., passenger flow). As a summary, studies related to these variables are reported through an approach of experiments and observation (see Table 1) to improve the LOS by reducing the dissatisfaction and delay time of passengers at the PTI, which is the main objective of this study.

**Table 1.** Variables that affect the passenger behavior in the PTI.

| Category | Variable | Reference | |
| --- | --- | --- | --- |
| | | Experiments | Observation |
| Physical | Door width (m) | Fernandez et al. [19]; Fujiyama et al. [20]; Fernandez et al. [21] | Harris [22]; Harris and Anderson [23]; Wiggenraad [24] |
| | Horizontal and vertical gap between the train and platform (mm) | Fernandez et al. [19]; Fujiyama et al. [20]; Daamen et al. [25]; Karekla and Tyler [26]; Seriani et al. [27] | Heinz [28]; Atkins [29] |
| | Steps between the train and platform (quantity) | Holloway et al. [30]; Seriani et al. [31] | Heinz [28]; Atkins [29] |
| | Platform width (m) | Seriani and Fernandez [4] | Harris [22]; Harris and Anderson [23] |

**Table 1.** *Cont.*

| Category | Variable | Reference | |
|---|---|---|---|
| | | Experiments | Observation |
| Spatial | Elevated platform (length, width, height) (m) | Tyler et al. [32] | Karekla et al. [33] |
| | Seats (quantity) | Fujiyama et al. [20] | Harris [22]; Harris and Anderson [23] |
| | Platform Edge Doors (quantity, width and type) | Seriani et al. [8]; De Ana Rodriguez et al. [34]; Seriani and Fujiyama [35]; Seriani and Fujiyama [36] | Seriani et al. [9]; Wu and Ma [37]; Loukaitou-Sideris et al. [38] |
| | Handrails, barriers, waiting area, yellow safety line (location, length, width) (m) | Seriani y Fernandez [4,39]; Valdivieso and Seriani [11]; Seriani et al. [18] | Wu and Ma [37]; Loukaitou-Sideris et al. [38] |
| Operational | Type of passenger (characteristics, luggage, reduced mobility, disability) | Seriani et al. [18]; Seriani et al. [27]; Holloway et al. [30] | Harris [22]; Harris y Anderson [23]; Wiggenraad [24]; Heinz [28]; Atkins [29]; Li et al. [40] |
| | Density (boarding passengers, alighting passengers, passengers remaining inside the train) (quantity or passengers/m$^2$) | Seriani et al. [8]; Seriani and Fujiyama [36]; Rowe and Tyler [41] | |
| | Space used by passengers (distance between passengers, area used by passengers) (m o m$^2$/passenger) | Seriani et al. [8]; Valdivieso and Seriani [11] | |
| | Distribution of passengers, formation of lines of flow, queues (quantity) | Seriani et al. [8]; Seriani and Fujiyama [36] | |
| | Flow of passengers at the doors (passengers/min-m) | Fujiyama et al. [20]; Fernandez et al. [21]; Daamen et al. [25] | |
| | Boarding and alighting time (s) | Fernandez et al. [19]; Holloway et al. [30]; De Ana Rodríguez et al. [34] | |

With respect to pedestrian simulation, there are two main approaches: macrosimulation and microsimulation [42]. The macrosimulation seeks to represent the movement of people as if it were a fluid. That is, the general behavior of the movement of pedestrians and their interactions with the surrounding environment are studied [43,44]. In the case of microsimulation, pedestrians are treated as individuals, where their characteristics and behavior are essential information in the representation and analysis of the simulation. These models can be classified into discrete, semicontinuous and continuous [45]:

- In discrete models, the variables involved are of the integer type. The best-known model is that of cellular automata. In the latter, the space where the simulation takes place is represented as a grid plane, where each cell represents a portion of the space with its respective rules [46]. In these models, the pedestrian is represented as a particle that moves from one cell to another and that can change state between each of these movements.
- In semicontinuous models, some variables are continuous, and others are discrete. For example, Guo [45] proposes a model in which the movement and space of pedestrians is continuous, and their location is updated at discrete time intervals.

- Continuous models use a domain in the real numbers for the variables involved and the entities move in a vector (continuous) manner. There are several models in this category, but the one of most interest in this study is the pedestrian dissatisfaction model LEGION, which is used in the current study. LEGION uses pedestrian dissatisfaction as the combination of three factors: inconvenience, frustration, and discomfort [4].

### 3. Simulation Method

*3.1. Variables and Scenarios*

There are different variables that influence the behavior of passengers in the PTI (see Table 1). Among them, those that are controllable by the designers are used in this study: the physical and spatial variables. In this case, it was decided to specifically evaluate how two variables affect the behavior of passengers: Platform configuration (or platform topology) and the use of traffic control elements (TCE).

With respect to the platform configuration, three scenarios were simulated:

1. Central (Center-Loaded Platform): This type of platform consists of a central platform located between the tracks for the trains. Passengers can board or alight trains in both directions from the platform. This is demonstrated in the following illustration (see Figure 1):

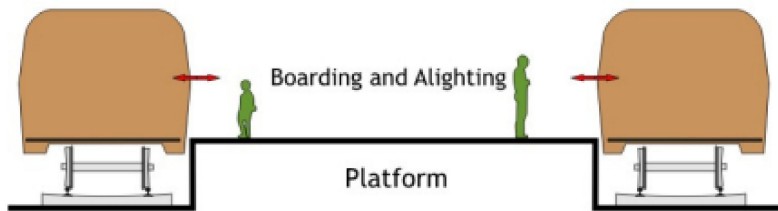

**Figure 1.** Central configuration of the platform (adapted from [47]).

2. Lateral (Side-Loaded Platform): This platform follows an inverse distribution to that of the central configuration, in which two platforms on the sides are located in the station, each one in a different direction (see Figure 2). In these, passengers must select which platform to board depending on the direction in which they wish to start their journey.

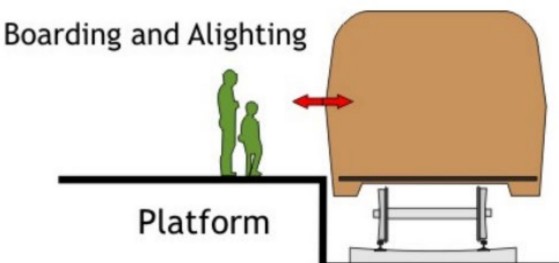

**Figure 2.** Lateral configuration of the platform (adapted from [47]).

3. Mixed (Flow-Through Platforms): Mixed platforms can be seen as the combination of the central and lateral platforms. They are the only ones that allow the use of all of the doors of the trains, in which three platforms are used (two exclusively for each train and one shared). This distribution can be seen in the following illustration (see Figure 3).

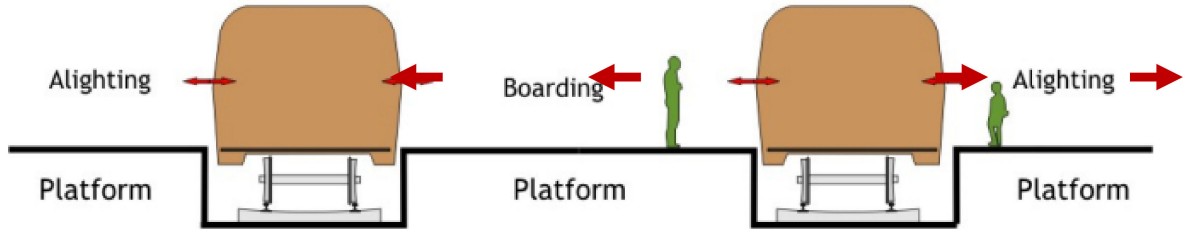

**Figure 3.** Mixed configuration of the platform (adapted from [47]).

In relation to TCE, different measures were used in the simulation:

- One-way doors: This TCE is based on the implementation of doors with an exclusive direction of entry/exit to/from the train towards the platforms in an interleaved manner. In this way, pedestrian counterflow at the doors is avoided. The direction of the induced behavior is demonstrated in the following illustration (see Figure 4).

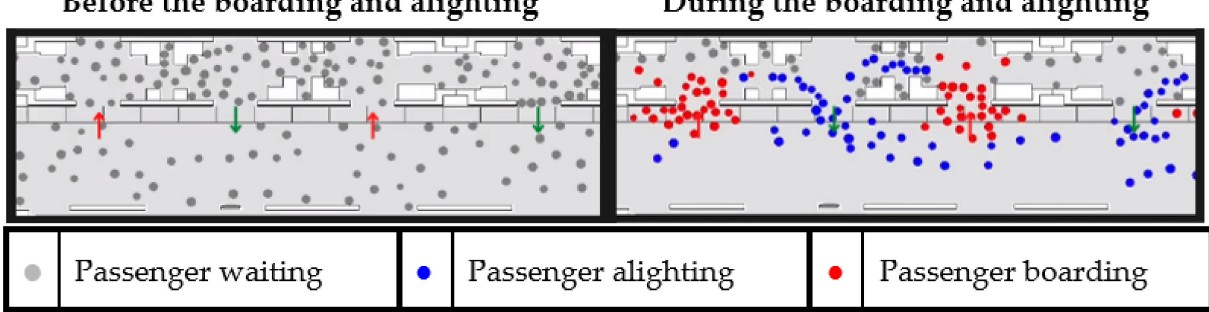

**Figure 4.** Use of one-way doors in the boarding and alighting process.

- Signage: The implementation of the "get off before getting on" behavior is based on the demarcation of the platform floor and indications that inform this behavior. Thus, pedestrian backflow at the doors is reduced. The following illustration shows its operation (see Figure 5).

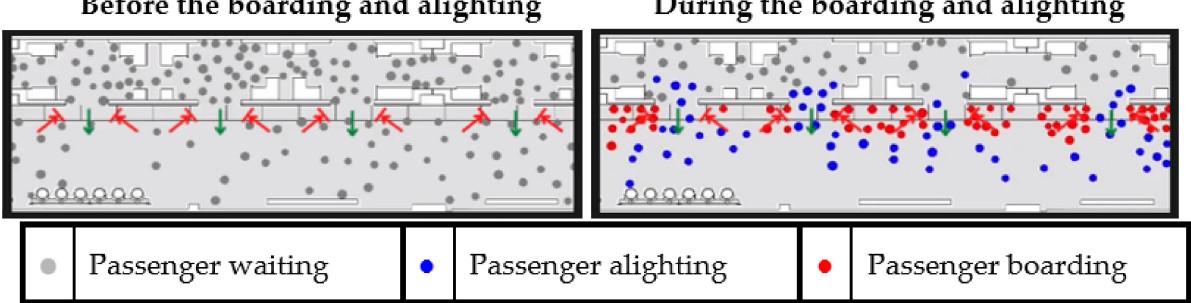

**Figure 5.** Use of signage "get off before getting on" in the boarding and alighting process.

In order to compare these types of platforms, the physical variables of the system are preserved. In other words, the total width of the waiting area on platforms, their length and the spatial distribution of exits are maintained without variation.

For the planimetry, a typical station of Santiago metro is used with some changes in its exits to represent a transit station. Disembarking and embarking passengers decide their route according to fair probabilities and proximity, respectively. The number of passengers used by the simulation comes from the extrapolation of the study by Seriani and Fernandez [4] from a single wagon to a complete train. The train model used in the study is NS 04, which is the typical train in the Santiago metro. The width of the platform

is 3.50 m and the length of the platform is about 130 m (corresponding to 9 cars of the train). In the case of the central platform, the configuration considered the tunnel as the same width, i.e., a platform width of 7.0 m (two times the platform width in the case of lateral Model). In addition, the platform included a yellow safety line at 0.6 m from the edge of the platform. In the case of a mixed Model (lateral and central platforms), each platform has a width of 2.3 m to allow a circulation of passengers boarding and alighting according to the design standards [7,12].

The behavior of the passengers in each scenario is the uncontrollable factor within the simulation. Therefore, it is sought to avoid its manipulation to generate legitimate results. This means that any behavior that is induced in passengers within the simulation is done indirectly.

### 3.2. Simulation Tool

In order to recreate the different PTI configurations, it was decided to use a pedestrian microsimulation tool as a modeling method. The tool selected was the LEGION simulator, which is mainly used and calibrated to simulate the boarding and alighting process in metro stations worldwide [2,7,48]. The objectives of using this tool are summarized in the following points:

- Model the pedestrian interchange in the PTI.
- Implement Control Traffic Elements to manage the passenger behavior.
- Collect output data that quantify the quality of station design, both quantitatively and qualitatively, to be subsequently analyzed using the factorial design method.

  To use LEGION, the following steps were required:

- Firstly, the input and output variables for the simulation were determined, along with the entities, spaces, and metrics necessary to measure the Fruin's LOS of each scenario.
- Secondly, the simulation scenarios were defined. As a base scenario, a typical station of line 1 of the Santiago Metro was used. Some adjustments were made to its design so that it represented accordingly to this study. A total of 9 combinations of the chosen variables were generated: two variables with three levels ($3^2$). The first variable corresponded to the platform configuration of the station (central, lateral, and mixed platforms), while the second variable referred to the use of traffic control elements (TCE) (Without TCE: one-way doors or signage "get off before getting on").
- Thirdly, once the scenario designs were finished, the simulations were executed through Legion Simulator. Sufficient repetitions were made for each simulation to achieve a correct modeling of the behavior function reflected in the metrics. Following this, the data of the performance metrics obtained from all the simulation scenarios were extracted. These were organized and sorted by TCE and platform configuration.
- Fourthly, in order to compare the performance of the different PTI configurations, an analysis was performed through a factorial design on the data obtained from the simulation, together with all the extra analyzes that could be extracted using the Minitab software. A factorial design of two factors and three levels (design $3^2$) were created, in which the implementation of 1 different TCE was evaluated in each analysis. The levels corresponded to lateral, central and mixed platforms. The factors were defined in relation to the TCE (one-way doors, signage "get off before getting on"). Subsequently, the results were analyzed for each TCE and for each platform configuration. The results were transferred into excel format files, reorganizing the data in the way preferred by the user, previously specified in the algorithm. For this purpose, KNIME Analytics was used to facilitate the subsequent transcription of the statistical data to Minitab.

  In the simulation, the following characteristics were defined:

- Simulation time: 00:01:30 [hh:mm:ss].
- Simulation events:

  (i)  Appearance of entities: 00:00:00 [hh:mm:ss].

(ii)　　　Start of exchange period: 00:00:10 [hh:mm:ss].

(iii)　　End of exchange period: instant of time in which the last entity existed in the PTI.

- Characteristics of the entities:

  (i)　　There was gender parity: 50% men and 50% women.

  (ii)　The entity profile was equivalent to that of a pedestrian in countries of southern Europe, given that their average speed is similar to the access speed for rail transport systems in Latin America. This speed is 5 km/h [12]

  (iii)　The use of luggage was not included in the experiment because it would imply an extra variable that would add high variability to the entire exchange process and may even have obscured the effect of the other factors.

  (iv)　Entity types:
    - Passenger inside the train: Entities inside the train that remain inside the train after the process of boarding and alighting.
    - Alighting passenger: Entities who alighted the train.
    - Boarding passenger: Entities who boarded the train.

- Scope of simulation:

  (i)　　The microsimulation model represented two simultaneous exchanges, one on each train line within the same station.

  (ii)　Exchanges with deterministic demand were generated within the platform, the same for all repetitions.

  (iii)　The boarding and alighting at the PTI was analyzed in detail with the chosen metrics, while the rest of the train and platform are generally observed.

To be able to run the simulations and collect the metrics of interest required the upload of the obtained LEGION Model Builder files (.LGM and .ORA) to LEGION Simulator. Then, in the Timeline configuration, the tables and heat maps based on the LOS of Fruin [10] were chosen. This provided average values within 0.6-s time intervals for the variables' dissatisfaction, entity density and social proximity counter. Finally, the simulator was configured to carry out 10 repetitions of the chosen model, which is equivalent to 20 train interchange repetitions, since each scenario was built based on the complete platform station (two trains).

### 3.3. KPIs and Indicators

The performance of each scenario could be quantified using indicators such as the LOS [10,12] and the dwell time [19]. Therefore, the time it takes for entities to get on and off the train at the opening and closing of doors could be obtained, as well as passenger satisfaction when using the metro service in the PTI (See Table 2).

**Table 2.** Level of service (LOS), density (pass/m$^2$) and space (m$^2$/pass).

| LOS | Values for Moving Passengers and Flat Areas | | | Values for Passengers in Waiting Areas | | Values for Both Cases |
|---|---|---|---|---|---|---|
| | Density $\left[\frac{pass}{m^2}\right]$ | Space $\left[\frac{m^2}{pass}\right]$ | Speed $\left[\frac{m}{s}\right]$ | Density $\left[\frac{pass}{m^2}\right]$ | Space $\left[\frac{m^2}{pass}\right]$ | Occupation[%] |
| A | ≤0.31 | ≥3.24 | ≥1.3 | ≤0.82 | ≥1.21 | 0–30 |
| B | 0.31–0.43 | 2.32–3.24 | 1.27–1.3 | 0.82–1.07 | 1.21–0.93 | 30–40 |
| C | 0.43–0.72 | 1.39–2.32 | 1.22–1.27 | 1.07–1.53 | 0.93–0.65 | 40–60 |
| D | 0.72–1.08 | 0.93–1.39 | 1.14–1.22 | 1.53–3.57 | 0.65–0.28 | 60–80 |
| E | 1.08–2.17 | 0.46–0.93 | 0.76–1.14 | 3.57–5.26 | 0.28–0.19 | 80–100 |
| F | ≥2.17 | ≤0.46 | ≤0.76 | ≥5.26 | ≤0.19 | ≥100 |

The KPIs used to measure the performance regarding the LOS and the dwell time in each scenario were defined from the Legion model [2,49] as follows:

- Average Boarding and Alighting Time (BAT): corresponds to the average time it takes for passengers to get on and off the train.
- Average Dissatisfaction: the average per person of a 'holistic' measure that quantifies the walking experience of each entity in comparison with their respective preferences; encompasses inconvenience, frustration, and discomfort. Pedestrians in Legion do their best to minimize their effort as it degrades the quality of their ride. Next, the nature of the three factors that comprise dissatisfaction is explained:
  - Inconvenience: the degree to which an entity must deviate from its preferred shortest distance.
  - Frustration: having to slow down in congested spaces (i.e., reduction in speed).
  - Discomfort: the perceived lack of adequate personal space.
- Average entity density: the average per person of the surrounding density. Entity density is roughly calculated for each passenger by drawing a 1.5 m-distance circular area around them, estimating the accessible space within it, and then estimating the number of entities within, including themselves. It is important to highlight that the 1.5 m is the distance around each passenger, i.e., it is the radius of a circular area.
- Average social proximity: the average of entities that infringe the personal space of a passenger during the simulation. In this study a 1-m radius is used.

From the data obtained in the outputs, the combination of the variables that optimized the Fruin's LOS and the performance of the station in operational terms was sought through factorial design.

The results were presented in the form of statistical graphs and data summary tables for each variable. In addition, heat maps were obtained based on the Fruin's LOS that they wanted to collect. This delivered average values within 0.6-s time intervals for the dissatisfaction, entity density, and social proximity counter variables.

The analysis performed a general factorial regression, composed of an ANOVA by metric ($\alpha = 0.05$), which yielded a series of tables and graphs depending on the options that were marked, which contained relevant information regarding the regression coefficients, the residuals, the effects of the factors and their interactions on the response variable. The results are presented in the next section.

The factorial design was performed using the analysis of variances (ANOVA), which generated a statistical quantification for each effect on the performance metrics. The analysis carried out by the factorial design also provided a series of comparative graphs on the effects, such as the Pareto graph or the semi-normal graph, which helped to demonstrate the contrast of each effect.

In addition, the Minitab's Response Optimizer was used to calculate an individual desirability using a desirability function (also called a utility transfer function). A weight from 0.1 to 10 was selected to determine how important it was to achieve the target value. The composite desirability was the weighted geometric mean of the individual desirability for the responses. To determine the optimal settings for the variables, Minitab maximized the composite desirability.

## 4. Results

### 4.1. Boarding and Alighting Time

The boarding and alighting time (or BAT) is the time it takes passengers to get on and off the train. It is made up of the boarding time and the alighting time. If the BAT is added to the time of opening and closing of doors and the time due to the acceleration and break of the train, then the dwell times are obtained.

The effect of the platform configuration and traffic control elements (TCE) on the BAT were analyzed. The null hypothesis s defined as the mean of the same being equal. The results show that the effect of the platform configuration factor is the most significant for

the BAT, which is reflected in the linear regression coefficients. However, in the case of TCE, the effect of one-way doors is not significant, having a *p*-value greater than 0.05 in the ANOVA. This means that both the use of one-way doors and its interaction with the platform configuration have almost no effect on BAT.

Using Minitab's Response Optimizer tool, the best treatment that minimized the time for passengers to get off and on was reached. The results were classified from the best to the worst performance according to the average BAT obtained in each scenario and desirability. The best scenario was the mixed model without one-way doors. It is important to note that Minitab's Response Optimizer calculates individual desirability using a desirability function (also called a utility transfer function). A weight (from 0.1 to 10) is selected to determine how important it is to achieve the target value. The composite desirability is the weighted geometric mean of the individual desirability for the responses. To determine the optimal settings for the variables, Minitab maximizes the composite desirability.

From Table 3 it can be concluded that the use of doors in one direction does not have a significant effect on the BAT. This means that it does not contribute to the improvement of the operational efficiency within the PTI. While the mixed platform configuration (Flow-Through Platforms) represents a clear improvement on the performance of the station, reducing the BAT by 6 s approximately.

**Table 3.** BAT considering one-way doors and different platform configurations.

| Ranking | Platform Configuration | Traffic Control Element | Average BAT (s) | Desirability |
|---------|------------------------|-------------------------|-----------------|--------------|
| 1 | Mixed | Without one-way doors | 15.18 | 0.825 |
| 2 | Mixed | With one-way doors | 16.31 | 0.757 |
| 3 | Lateral | Without one-way doors | 21.60 | 0.436 |
| 4 | Central | With one-way doors | 21.93 | 0.416 |
| 5 | Lateral | With one-way doors | 22.11 | 0.405 |
| 6 | Central | Without one-way doors | 22.53 | 0.380 |

In the case of the TCE signage "get off before getting on", its effect depends on each platform's configuration. In the lateral configuration, a large increase in BAT is seen when using signage, while in the central configuration it is slightly lower and in the mixed configuration there is no significant variation. This is due to the fact that, unlike the lateral and central type of platforms, the mixed platform has the advantage of having twice as many doors and platforms for the boarding and alighting of passengers, and thus allows the demand of entities per door to be distributed and a less competitive scenario is generated by going through the doors. This gives the possibility of having greater freedom of movement and reducing the dissatisfaction for people. Therefore, the effect of the signage is not noticeable in the BAT as in the other platform topologies, since in the mixed platform without signage people have the possibility of crossing doors without close contact with other pedestrians, similar to what happens in the scenario with signage. Therefore, the model for mixed platform with signage is ranked first, mainly due to the effect of the platform configuration factor, where, as explained above, a better use is made of the space available on the platforms. In addition, the signage allows the induction of the "get off before getting on" behavior, which reduces the counterflow of passengers, allowing a more controlled boarding and alighting, without obstacles. The two combinations that were found to best minimize BAT are the mixed platform configuration with and without signage (See Table 4).

**Table 4.** BAT considering signage "get off before getting on" and different platform configurations.

| Ranking | Platform Configuration | Traffic Control Element | Average BAT (s) | Desirability |
|---|---|---|---|---|
| 1 | Mixed | With signage | 15.15 | 0.862 |
| 2 | Mixed | Without signage | 15.18 | 0.860 |
| 3 | Lateral | Without signage | 21.60 | 0.550 |
| 4 | Central | Without signage | 22.53 | 0.505 |
| 5 | Central | With signage | 26.25 | 0.326 |
| 6 | Lateral | With signage | 26.64 | 0.307 |

In the case of lateral platform configuration, the use of signage increased the BAT by approximately 5 s. This is due to the fact that during the interchange phase without signage, counterflows force pedestrians to concentrate more on the cross section of the door, struggling to walk towards their points of interest and creating microflows in opposite directions, reaching a higher density. If the signage is implemented, only one line of flow is formed, reaching a lower density (see Figure 6). The reason for this is because people do not need to compete to be able to pass through the door, being able to do it in a more satisfactory way by keeping a distance from other pedestrians.

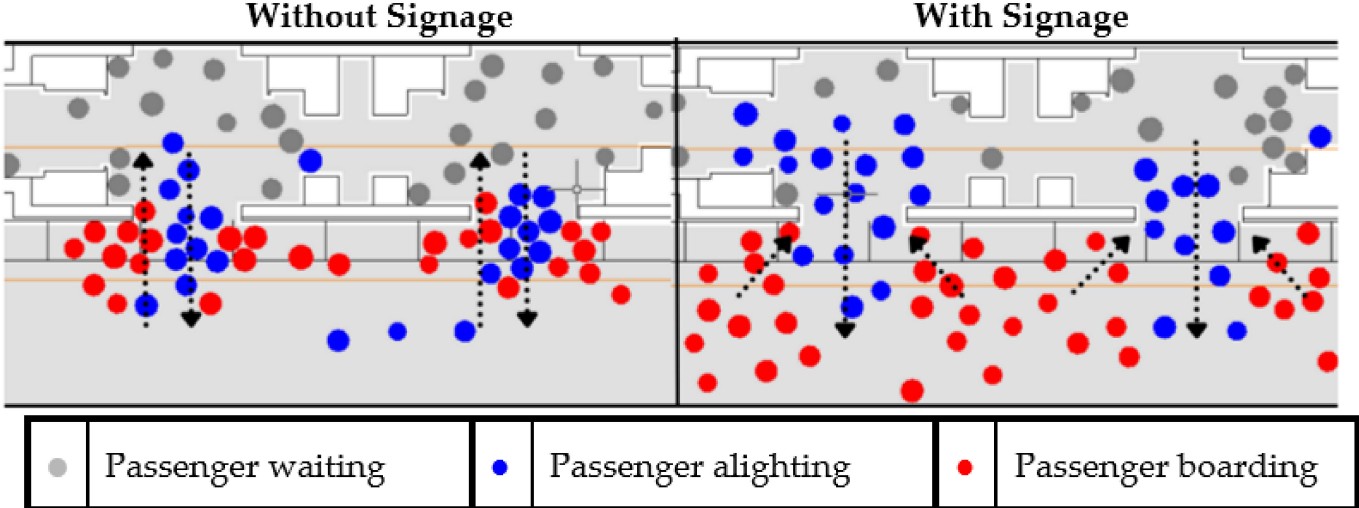

**Figure 6.** Use of signage "get off before getting on" in the boarding and alighting process when using a lateral platform configuration.

The simulation design consists of a 3 × 2 type, which generates a non-linear regression by using interactions, so the predictions made do not necessarily have to be fulfilled, but it is very useful to analyze the behavior of the variables and their comparability. Here are some indications:

-   The case of one-way doors has an adjusted R square of 57.49%, which indicates a sufficient percentage to explain the variables. In the case of signage as a traffic control element, an R square equal to 78.42% is reached.
-   The residuals do not stray that far from the fitted values.
-   The points on the normal probability plot form a line.
-   The histogram of residuals approaches a normal distribution.

### 4.2. Dissatisfaction

As it is mentioned in Section 3, dissatisfaction is a "holistic" measure of each entity's walking experience compared to their respective preferences; it encompasses inconvenience,

frustration and discomfort. It is an indicator that represents the extent to which entities fail to meet their mobility and transportation needs. Therefore, the model that manages to minimize this indicator will be the best in terms of service for passengers.

The use of one-way doors had no stable incidence on the results of dissatisfaction for all the platform configurations. In Table 5, the interaction of the use of one-way doors with respect to the platform configuration is captured. In it is observed a similar behavior to that observed in the central and lateral models, lowering their average dissatisfaction by approximately 1 unit. In contrast, the complete opposite is observed in the mixed model, in which an average dissatisfaction increased. This contradictory and erratic behavior is the reason why a significant effect cannot be attributed to the use of one-way doors. This is explained by the same reasons raised in the analysis when comparing heat maps. It must be remembered that entity density and dissatisfaction are directly related due to discomfort. The central model presents slightly higher dissatisfaction than the lateral model, both with and without one-way doors. This has to do with the distribution of outputs. As there were only exits at the extreme right and left in the model with a central platform, pedestrians formed groups towards the direction closest to them. This effect was further enhanced with the one-way door, generating greater agglomeration in exchange for the elimination of the backflow.

**Table 5.** Dissatisfaction considering one-way doors and different platform configurations.

| Ranking | Platform Configuration | Traffic Control Element | Average Dissatisfaction (Unitless) | Desirability |
|---|---|---|---|---|
| 1 | Mixed | Without one-way doors | 8.06 | 0.914 |
| 2 | Lateral | With one-way doors | 9.63 | 0.584 |
| 3 | Mixed | With one-way doors | 10.18 | 0.467 |
| 4 | Central | With one-way doors | 10.38 | 0.424 |
| 5 | Lateral | Without one-way doors | 10.78 | 0.342 |
| 6 | Central | Without one-way doors | 11.36 | 0.219 |

In relation to the ANOVA (see Table 5), the *p*-value indicates how accurate the variables are. The *p*-value for the use of one-way doors is too large, making it insignificant for the model. Using Minitab's Response Optimizer, we seek to find the combinations that minimize dissatisfaction. The result found was a list of solutions from the best to the worst, according to the average dissatisfaction obtained in each one and their composite desirability. The best of the solutions found was the mixed model without ne-way doors.

It is concluded that the use of one-way doors does not have a significant effect on dissatisfaction; however, by interacting with the platform configurations, it generates different changes, slightly improving the lateral and central platforms. This means that it does contribute to improving the level of service for passengers in both models. On the other hand, the mixed platform configuration increases the average dissatisfaction when using one-way doors. Its contribution to other aspects is not ruled out and the use of one-way doors with a strict approach on the central platform and alighting on the sides is pending another study.

In relation to the signage "get off before getting on", it was found to reach the highest reduction in the dissatisfaction. This is understandable, since the lateral and central platforms both occupy only half of their capacity in doors, reducing the greatest difference between these models in their arrangement of space on the platform. However, the mixed platform model manages the greatest reduction in people's dissatisfaction by having the option of distributing the exchange demand in twice the number of doors than the other two platform configurations.

Regarding the interactions between factors (see Table 6), it is observed that the signage "get off before getting on" reduced the dissatisfaction of passengers in all platform configurations in a very similar way, reaching the maximum reduction in the case of mixed platforms. Therefore, this means that it is universally applicable in all types of metro stations to reduce dissatisfaction. The effect of the interaction between signage and the central and lateral platform configurations is striking. It can be seen that without signage, the lateral platform reduced dissatisfaction more than the central platform, but in a scenario with signage, the central platform reduced dissatisfaction more than the lateral platforms. In other words, the signage had more of an effect on the central platform than on the lateral configuration of the platforms. Furthermore, the decision to implement a central or lateral platform depends on whether the use of signage will be considered or not. It is worth mentioning the possibility that the effect of the signage on dissatisfaction is related to the width of the platforms, which remains a topic for future research.

**Table 6.** Dissatisfaction considering signage "get off before getting on" and different platform configurations.

| Ranking | Platform Configuration | Traffic Control Element | Average Dissatisfaction (Unitless) | Desirability |
|---|---|---|---|---|
| 1 | Mixed | With signage | 7.08 | 0.913 |
| 2 | Mixed | Without signage | 8.06 | 0.745 |
| 3 | Central | With signage | 9.58 | 0.484 |
| 4 | Lateral | With signage | 9.71 | 0.460 |
| 5 | Lateral | Without signage | 10.77 | 0.287 |
| 6 | Central | Without signage | 11.35 | 0.178 |

Finally, the simulation design consisted of a $3 \times 2$ type, which generated a non-linear regression by using interactions. Here are some indications:

- The case of one-way doors had an adjusted R square of 75.79%, which indicates a sufficient percentage to explain the variables. In the case of signage as a traffic control element, an R square equal to 90.84% was reached.
- The residuals do not stray that far from the fitted values.
- The points on the normal probability plot form a line.
- The histogram of residuals approaches a normal distribution.

### 4.3. Entity Density

Entity Density is a metric used in LEGION Simulator and corresponds to the roughly calculated density for each entity by drawing a 1.5 m circular area around it, estimating the accessible space within that area, and then estimating the number of entities within the area, including the entity itself.

As predicted, in the interaction for all platform configurations experienced an increase in their entity density when using one-way doors. Despite being able to efficiently eliminate backflow, the use of one-way doors increased the crowding of passengers to getting out or getting in the train, even before the doors opened. As can be seen, the increases were from 2.2 to 2.6 (lateral platform), 2.3 to 2.6 (central platform) and 2.2 to 3.0 (mixed platform). This means an average increase of 0.4 people within the personal area of each passenger. Observing this information, there is evidence of a substantial increase in entity density in the mixed platform compared to the other two configurations (lateral and central platform). Although at first glance the opposite might be expected, if the high-density maps of the mixed models with and without one-way doors are compared (see Figure 7), a relevant difference is seen in the interchange area and inside the train. Blue represents a low level of entity density while red represents a high level of entity density. In the model with

one-way doors, higher densities are seen, especially inside the train. This makes sense because the congestion in this model, in general, is greater by reducing the exit/entry options of pedestrians by half, which generates less dispersion among them on the way to the doors; this, in turn, increases the formation of groups and the collision between passengers inside the train due to having to travel a longer distance. The main objective of one-way doors is to eliminate the backflow that occurs when boarding and alighting. However, in the case of the mixed platform, the counterflow that exists with this number of people is somewhat low, and instead there is a greater crowding effect inside the train due to the need to transfer to another car after boarding.

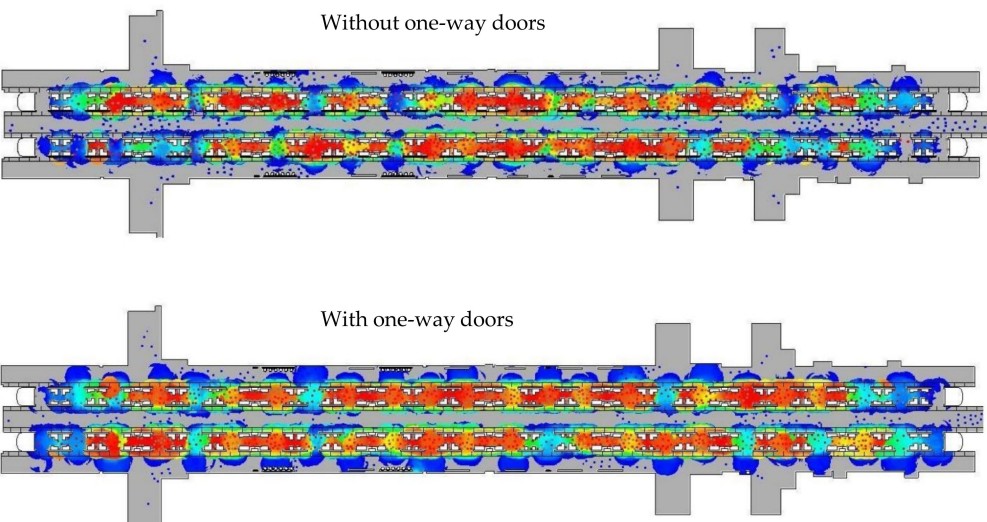

**Figure 7.** Map of high entity densities in mixed platform with and without the implementation of one-way doors.

The central model had a slightly higher density than the lateral model, both with and without one-way doors. This has to do with the distribution of outputs and the existence of two platforms vs. one platform. As there were only exits on the extreme right and left in the model with a central platform, pedestrians formed groups towards the closest destination. This effect was enhanced by the implementation of one-way doors, generating greater agglomeration in exchange for the elimination of backflow.

Using Minitab's Response Optimizer tool (see Table 7), we sought to find the best combination that minimized entity density. The result found was a list of solutions from the best to the worst according to the average entity density obtained in each one, and according to their composite desirability. The best solution was the mixed model without one-way doors. It is concluded that the use of one-way doors does have a significant effect on entity density, but tends to increase it in all models, without exception. This means that it does not contribute to improving the level of service for passengers. Its contribution to other aspects is not ruled out, which may be studied in further simulations.

The signage "get off before getting on" greatly affected and minimized the density of entities at its high level. Implementing the "get off before boarding" behavior allowed boarding passengers to be spread out on the sides of the doors while passengers inside the train can get off unhindered. In this way, a better use of space was made by distributing the entities and avoiding backflows that saturated the door with their stops.

In addition, it can be seen that the implementation of signage decreased entity density in all platform configurations, but not always in the same way, as it had a greater effect on lateral and central platforms compared with mixed platforms.

**Table 7.** Entity Density considering one-way doors and different platform configurations.

| Ranking | Platform Configuration | Traffic Control Element | Average Entity Density (Passengers) | Desirability |
|---------|------------------------|-------------------------|-------------------------------------|--------------|
| 1 | Mixed | Without one-way doors | 2.16 | 0.875 |
| 2 | Lateral | Without one-way doors | 2.24 | 0.807 |
| 3 | Central | Without one-way doors | 2.28 | 0.764 |
| 4 | Lateral | With one-way doors | 2.66 | 0.414 |
| 5 | Central | With one-way doors | 2.66 | 0.408 |
| 6 | Mixed | With one-way doors | 2.99 | 0.104 |

The effect of the signage on the platform configurations made the mixed platform station go from being the best to the least practical to reduce the entity density, although with minimal differences. People who board and wait at the sides of the doors tend to be closer together near the doors and more spread out at the end of the line to enter. This causes a higher density in mixed platforms, since more short queues are created (closer to the door) than on lateral and central platforms.

The lateral platform with signage is the most prepared configuration to reduce the density of entities in the PTI. This is because it completely separates the demand; therefore, it is practical for symmetrical flows, where the demand of one train does not interact with the other. It is the PTI configuration in which the number of people waiting to board is also distributed in the platform waiting area, creating points of high density, but behind the yellow safety line in front of the train doors.

The Minitab's Response Optimizer tool was used to deliver the best possible solutions (see Table 8). As can be seen, the order of the best PTI configurations was highly affected in a scenario with and without signage. All the scenarios with signage managed to work better than the entity density than the scenarios without signage, corroborating the effectiveness of implementing the behavior "get off before getting on".

**Table 8.** Entity Density considering signage "get off before getting on" and different platform configurations.

| Ranking | Platform Configuration | Traffic Control Element | Average Entity Density (Passengers) | Desirability |
|---------|------------------------|-------------------------|-------------------------------------|--------------|
| 1 | Lateral | With signage | 1.83 | 0.911 |
| 2 | Central | With signage | 1.84 | 0.886 |
| 3 | Mixed | With signage | 1.88 | 0.817 |
| 4 | Mixed | Without signage | 2.16 | 0.340 |
| 5 | Lateral | Without signage | 2.24 | 0.218 |
| 6 | Central | Without signage | 2.28 | 0.139 |

Finally, the simulation design consists of a $3 \times 2$ type, which generates a non-linear regression by using interactions. Here are some indications:

- The case of one-way doors had an adjusted R square of 94.91%, which indicates a sufficient percentage to explain the variables. In the case of signage as a traffic control element, an R square equal to 93.71% was reached.
- The residuals did not stray that far from the fitted values.
- The points on the normal probability plot formed a line.
- The histogram of residuals approached a normal distribution.

*4.4. Social Proximity*

According to the Legion user manual, an entity's proximity count is the number of entities that are within its social distance (defined as 1 m), including itself.

The difference that the use of one-way doors had between the models is remarkable. This means that the effect of this factor on the proximity count is much more significant than the platform configuration, which is reflected in its linear regression coefficient. Note also that there is an equivalence between the effects of the interaction and that of the platform configuration. This indicates that the effect of the interaction between the factor is applied with the same magnitude in the three configurations of platforms.

As predicted, all platform configurations showed an increase in social proximity when using one-way doors. It is worth noting that this increase in social proximity was quite low for the lateral and central models. The mixed platform did undergo a greater change, for the same reason explained in the entity density analysis in Section 4.3. There was an increase in pedestrian congestion when using one-way doors, and this exceeded the counterflow that seeks to eliminate the use of them, since even without them, pedestrians follow an intuitive behavior of giving importance to those who alight or board the train. Specifically in the mixed model, this is accentuated by having twice the number of exits and entrances (half the demand per door in one direction). The situation is repeated again in the central model, which presents a slightly higher density than the lateral model, both with and without one-way doors. This has to do with the distribution of outputs. As there are only exits on the right and left ends in the model with a central platform, pedestrians formed groups towards the closest destination (only two). This effect was enhanced with the one-way doors, again, generating greater accumulation in return backflow elimination.

Using Minitab's Response Optimizer tool (see Table 9), we sought to find the best treatment that minimized the proximity count. The mixed model without using one-way doors is the most beneficial solution in this metric. It was concluded that the use of one-way doors does have a significant effect on the proximity count, but it tends to increase, especially in the mixed model. This means that it does not contribute to improving the level of service for passengers. Its contribution to other aspects is not ruled out, which may be studied in further simulations.

**Table 9.** Social proximity considering one-way doors and different platform configurations.

| Ranking | Platform Configuration | Traffic Control Element | Average Social Proximity (Passengers) | Desirability |
|---|---|---|---|---|
| 1 | Mixed | Without one-way doors | 5.18 | 0.923 |
| 2 | Lateral | Without one-way doors | 6.35 | 0.423 |
| 3 | Central | Without one-way doors | 6.49 | 0.363 |
| 4 | Lateral | With one-way doors | 6.54 | 0.343 |
| 5 | Mixed | With one-way doors | 6.66 | 0.288 |
| 6 | Central | With one-way doors | 6.77 | 0.244 |

In relation to the "get off before getting on" behavior, the use of this signage effectively reduced the number of interactions at all levels of the platform configurations. Its effect was more evident in stations with a platform design that only allowed the use of a row of train doors for interchange, being slightly greater in stations with a central platform. The effect in the mixed platform model was considerably less than in the other platform topologies, but still significant. The different platform configurations showed different performances in the number of interactions when the signage was used or when it was not. The central configuration was the least optimal, followed by the lateral platform. The most effective was the mixed platform. In particular, it is striking how the interactions between passengers changed when signaling is applied to the stations. In the case of signage, not

only did close encounters between pedestrians decrease, but also the number they were reduced to is practically the same. This is a great point in favor of the factor that induces the "get off before getting on" behavior, since it shows that it makes any type of station just as efficient in the case of needing to reduce interactions between passengers, as it is in a context of pandemic.

As analyzed above, it makes sense that the best combination of levels to minimize close encounters between passengers is the mixed platform configuration with signage. This configuration has the advantage of occupying all the doors of the train. In this way, the number of passengers who are interested in going through a specific door is reduced compared to stations with lateral platforms and central platforms, making interactions between pedestrians less likely. In addition, allowing people to alight before boarding the train also promotes a distance between people by reducing backflows. These reasons make the mixed model the ideal configuration to ensure a safe exchange between passengers.

The Minitab's Response Optimizer tool is used to deliver the best possible solutions (see Table 10). As can be seen, the best configuration is the mixed model in which the average social proximity is highly reduced. All the scenarios with signage managed to work better in terms of social proximity than the scenarios without signage, corroborating the effectiveness of implementing the behavior "get off before getting on".

**Table 10.** Social proximity considering signage "get off before getting on" and different platform configurations.

| Ranking | Platform Configuration | Traffic Control Element | Average Social Proximity (Passengers) | Desirability |
|---|---|---|---|---|
| 1 | Mixed | With signage | 4.49 | 0.923 |
| 2 | Central | With signage | 4.58 | 0.892 |
| 3 | Lateral | With signage | 4.58 | 0.889 |
| 4 | Mixed | Without signage | 5.18 | 0.666 |
| 5 | Lateral | Without signage | 6.34 | 0.230 |
| 6 | Central | Without signage | 6.48 | 0.178 |

Finally, the simulation design consists of a $3 \times 2$ type, which generates a non-linear regression by using interactions. Here are some indications:

- The case of one-way doors had an adjusted R square of 94.91%, which indicates a sufficient percentage to explain the variables. In the case of signage as a traffic control element reach an R square equal to 97.41%.
- The residuals did not stray that far from the fitted values.
- The points on the normal probability plot formed a line.
- The histogram of residuals approached a normal distribution.

## 5. Discussion: Global Optimization

Taking into account the analyses for the use of one-way doors and for the use of signage ("get off before getting on"), a final comparison was made, this time taking into account both traffic control elements and the platform configurations, with the purpose of finding the most effective solution. Therefore, a factorial design of two factors and three levels ($3^2$) was used as a corroboration method for the resolutions, applying Minitab analyzer in order to find an optimal response for each of the following cases.

The passenger boarding and alighting time is a KPI that helps to measure the performance of a station in the platform-train interface (PTI), in terms of speed of operation. According to what happened in the previous analyses, the results are quite expected. The best model turned out to be the mixed platform with signage, and in second place, the mixed platform without signage or one-way doors. Lastly, the lateral platform without

signage or one-way doors is quite well positioned at position number 4 (see Table 11). It is important to note that for this metric, the models that have a mixed platform configuration are positioned in the first three positions, which demonstrates high operational efficiency in relation to the rest.

**Table 11.** Optimal response to the BAT considering a design of experiment $3^2$.

| Ranking | Platform Configuration | Traffic Control Element | Average BAT (s) | Desirability |
|---|---|---|---|---|
| 1 | Mixed | With signage | 15.15 | 0.862 |
| 2 | Mixed | Without signage or one-way doors | 15.18 | 0.860 |
| 3 | Mixed | With one-way doors | 16.31 | 0.806 |
| 4 | Lateral | Without signage or one-way doors | 21.6 | 0.550 |
| 5 | Central | With one-way doors | 21.93 | 0.534 |
| 6 | Lateral | With one-way doors | 22.11 | 0.526 |
| 7 | Central | Without signage or one-way doors | 22.53 | 0.505 |
| 8 | Central | With signage | 26.25 | 0.326 |
| 9 | Lateral | With signage | 26.64 | 0.307 |

With respect to dissatisfaction and entity density, these two KPIs represent the level of service in the PTI. This means that the best model represents the ideal to increase satisfaction and reduce the risk of passengers in the PTI. As expected, the best solution for multiple optimal response is the mixed platform with signage, followed by the mixed platform without signage or one-way doors. However, they are closely followed, even with a lower density, by the central platform with signage and the lateral platform with signage (see Table 12). While the lateral platform without signage or one-way doors is positioned at number 5, with a larger gap compared to the models of the previous positions.

**Table 12.** Optimal response to the dissatisfaction and entity density considering a design of experiment $3^2$.

| Ranking | Platform Configuration | Traffic Control Element | Average Entity Density (Passengers) | Average Dissatisfaction (Unitless) | Desirability | Fruin's Level of Service |
|---|---|---|---|---|---|---|
| 1 | Mixed | With signage | 1.88 | 7.08 | 0.916 | E |
| 2 | Mixed | Without signage or one-way doors | 2.16 | 8.06 | 0.726 | E |
| 3 | Central | With signage | 1.84 | 9.58 | 0.678 | E |
| 4 | Lateral | With signage | 1.83 | 9.71 | 0.665 | E |
| 5 | Lateral | Without signage or one-way doors | 2.24 | 10.77 | 0.426 | F |
| 6 | Lateral | With one-way doors | 2.66 | 9.62 | 0.399 | F |
| 7 | Central | With one-way doors | 2.66 | 10.38 | 0.337 | F |
| 8 | Central | Without signage or one-way doors | 2.28 | 11.35 | 0.332 | F |
| 9 | Mixed | With one-way doors | 2.99 | 10.18 | 0.178 | F |

From Table 12, it can be seen that most of the cases have a level of service (LOS) that exceeds the standard in stations. This occurs due to the high demand entity density that is studied in each case. The most used configuration in Chile is the lateral platform without

traffic control elements, which is positioned right in the middle of Table 12, with a LOS = F. In addition, as can be seen in Table 12, all platform configurations with signage have a better LOS than their counterparts without signage or with one-way doors. In the case of the lateral platform, the behavior "get off before getting on" reached a better LOS than the case without signage. In addition, two of the models with a mixed platform configuration reflected a more convenient LOS than the rest of the cases. This reaffirms that the best situation is offered by the mixed platforms. Regarding the implementation of one-way doors, it was found that although they induce a reduction in pedestrian counterflows, they is not effective in mitigating the entity density (they can even increase it).

With respect to social proximity, one of the difficulties that people have faced in respecting self-care measures against COVID-19 is the ability to maintain social distancing on public transport. That is why this metric is given importance, to identify which platform configuration is the one that best adapts to the needs of having fewer interactions between passengers. However, it is important to mention that even if the number of interactions is reduced in a high-demand scenario, this does not mean that passenger safety is guaranteed as it only represents a reduction in the probability of risk. The possible solutions are presented in Table 13, in which the most effective case to reduce interactions is the mixed platform configuration model with signage, followed by the central platform with signage, and, in third place, the lateral platform model with signage.

**Table 13.** Optimal response to the social proximity considering a design of experiment $3^2$.

| Ranking | Platform Configuration | Traffic Control Element | Average Social Proximity (Passengers) | Desirability |
|---------|------------------------|--------------------------|----------------------------------------|--------------|
| 1 | Mixed | With signage | 4.49 | 0.933 |
| 2 | Central | With signage | 4.58 | 0.905 |
| 3 | Lateral | With signage | 4.58 | 0.902 |
| 4 | Mixed | Without signage or one-way doors | 5.18 | 0.706 |
| 5 | Central | Without signage or one-way doors | 6.34 | 0.324 |
| 6 | Lateral | Without signage or one-way doors | 6.48 | 0.278 |
| 7 | Lateral | With one-way doors | 6.53 | 0.262 |
| 8 | Mixed | With one-way doors | 6.66 | 0.221 |
| 9 | Central | With one-way doors | 6.76 | 0.187 |

Finally, the configuration that is most prepared to minimize the chosen metrics, and thereby optimize the operational performance of the station and its level of service, is the mixed configuration platform with signage. The second configuration that has the best performance is the mixed platform without signage or one-way doors, and the third on the list is the central platform design with signage (see Table 14).

**Table 14.** Optimal response to the BAT, dissatisfaction, entity density and social proximity, considering a design of experiment $3^2$.

| Ranking | Platform Configuration | Traffic Control Element | Average BAT (s) | Average Social Proximity (Passengers) | Average Entity Density (Passenger) | Average Dissatisfaction (Unitless) | Desirability |
|---|---|---|---|---|---|---|---|
| 1 | Mixed | With signage | 15.15 | 4.49 | 1.88 | 7.08 | 0.906 |
| 2 | Mixed | Without signage or one-way doors | 15.18 | 5.18 | 2.16 | 8.06 | 0.752 |
| 3 | Central | With signage | 26.25 | 4.58 | 1.84 | 9.58 | 0.606 |
| 4 | Lateral | With signage | 26.54 | 4.58 | 1.83 | 9.71 | 0.591 |
| 5 | Lateral | Without signage or one-way doors | 21.60 | 6.34 | 2.24 | 10.77 | 0.424 |
| 6 | Lateral | With one-way doors | 22.11 | 6.53 | 2.66 | 9.62 | 0.385 |
| 7 | Central | Without signage or one-way doors | 22.53 | 6.48 | 2.28 | 11.35 | 0.353 |
| 8 | Central | With one-way doors | 21.93 | 6.76 | 2.66 | 10.38 | 0.326 |
| 9 | Mixed | With one-way doors | 16.31 | 6.66 | 2.99 | 10.18 | 0.274 |

## 6. Conclusions

The objective of this research is to search for a model that meets the different needs of passengers with respect to the metrics of interest, to optimize the operational performance and level of service (LOS) in the station, specifically the platform-train interface (PTI).

When analyzing the results, three important verdicts were reached regarding the comparison of the models and considerations to take into account.

First, the use of one-way doors is a factor that has a notoriously weak and unfavorable effect in most cases for the metrics of interest, which leaves little to say about it in terms of results. However, this does not mean that it is a useless model in other cases. In fact, it is capable of reducing passenger dissatisfaction when using central and lateral platforms.

When extrapolating to a scenario with a mixed platform configuration, but restricting the approach to the central platform and the lateral platform configurations, a flow-through platform type model is generated. In these models, a unidirectional flow is generated for boarding and alighting, avoiding counterflow and ordering the route carried out by passengers. The formation of lines of flow were not carried out due to lack of time, because it involved going beyond the scope of this study as it required the creation of other scenarios that were outside the used configuration and design of the PTI. Thus, an experimental analysis that includes this scenario is pending for future research.

Second, in contrast to the above, the use of signage has a high positive impact in minimizing most of the metrics of interest for all platform configurations. For this reason, in global optimization, it is presented as the best solution. In terms of global optimization, it can be considered that the mixed platform with signage is the best option in terms of operating performance, passenger satisfaction and social distancing as a measure of personal care, followed by the model of mixed platforms without use of traffic control elements.

However, the mixed platform model may have additional requirements that the other types of platform configuration do not have. For example, the mixed platform requires the expansion of the width of the tunnel and restructuring of the entrances and exits of the station. Likewise, from an operational point of view, the boarding and alighting time (BAT) obtained by the lateral model without traffic control elements does not present a bad prospect if it is desired to maintain it to reduce expenses.

With respect to the level of service, the central platform model with signage and the lateral platforms with signage have a performance comparable to the mixed platform model. The same goes for the social distancing. Apart from the lateral platform configuration, the use of signage that induces "get off before getting on" behavior in these facilities is

strongly recommended. This is a fairly inexpensive option that can optimize all metrics except the BAT.

The advantages of the mixed platform model over the central and lateral configurations with and without signage are demonstrated. The strengths of the implementation of one-way doors are associated with the safety of pedestrians when they travel through the circulation spaces, since they reduce the interaction between passengers. Unfortunately, including this traffic control element greatly harms the operational variables of the simulated system.

The challenge remains to evaluate the PTI configurations in other asymmetric exchange scenarios in order to study their performance in different demand conditions and platform widths. In addition, it remains to evaluate the other scenario mentioned above with a mixed platform configuration and one-way doors, which alters the direction of the use of doors. Finally, it would be interesting to study the effect of moving passengers inside the train and to find other ways to change the PTI space, so that it brings benefits for the boarding and alighting of passengers, considering people with reduced mobility or disabilities.

**Author Contributions:** Conceptualization, T.F.; methodology, M.K., A.W. and S.S.; software, M.K. and A.W.; validation, M.K., A.W. and S.S.; formal analysis, M.K. and A.W.; investigation, S.S.; resources, S.S.; data curation, M.K. and A.W.; writing—original draft preparation, M.K. and A.W.; writing—review and editing, S.S. and T.F.; visualization, M.K. and A.W.; supervision, T.F.; project administration, S.S.; funding acquisition, S.S. All authors have read and agreed to the published version of the manuscript.

**Funding:** This study was financed by FONDECYT project 11200012 "Passenger on Urban Railway Platforms by Laboratory Experiments (PURPLE)", ANID, Chile.

**Institutional Review Board Statement:** Not applicable.

**Informed Consent Statement:** Not applicable.

**Data Availability Statement:** Not applicable.

**Conflicts of Interest:** The authors declare no conflict of interest.

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
