# Peer review of "Factorial Design with Simulation for the Optimization of the Level of Service in the Platform-Train Interface of Metro Stations—A Pilot Study"

_sustainability, doi:10.3390/su142315840_

Round 1
Reviewer 1 Report
The presented problem is important and this study is constructed well in general. However, I indicated some problems (especially in the description of the research) that should be solved before the second round of the review.
The arrows in figure 3 should correspond with the assumed direction of boarding and alighting (arrows with only one head).
Section 3.2 contains the description of the used simulation tool. I think that the classification of models is not necessary there, thus I recommend deleting rows 249 – 271.
Row 313, have females or males different behaviors in the used simulation tool? What is a sense of indication of gender parity (50/50)? Will other proportions of parities influence the results?
I have the remarks on the construction of tables 3 – 10. Columns entitled “solution” suggest some kind of configuration of platform and doors as 1 – 6. This is confusing because an example: “solution 2” from table 3 means “mixed platform configuration” and a use of “one-way doors”, whereas “solution 2” from table 5 means “lateral platform configuration” and a use of “one-way doors”, “solution 2” from table 7 means “lateral platform configuration” and a lack of “one-way doors” etc.
In fact, the tables present some ranking considering specific criteria. It needs to be clarified. Considered solutions (named "options" or "models") should be better defined and classified.
If tables 3 – 10 present the order of evaluated solutions, the first column can be renamed into “ranking”. Another way is some presentation of solutions for example option 1 = “mixed platform configuration” and use of “one-way doors”, option 2 = “mixed platform configuration” and a lack of “one-way doors” etc.
The next aspect is dedicated to a code for the occurrence or lack of “one-way doors” or “signage”. I mean, the use of “+1” or “-1” is too complex. The better way will be to write: “yes” or “no”.
The manuscript should contain the next section called “discussion” situated before the last section “conclusions”. This new section should contain a comparison of obtained results with other similar studies (known from the literature).
Last question: did the Authors model the movement of passengers in the trains when the metro runs between the stations? How influence the solutions (whit different organizations of doors, platforms, and signage) on passengers' behaviors? How does it influence LOS identification?
Author Response
Thank you very much for the comments. We tried to address all of them as best as possible. Please find attached the responses.

Author Response

(The authors gave the same response as above.)

Round 2
Reviewer 1 Report
I am reporting that all of my remarks were successfully considered. Now, the paper is more clear and readable. In my opinion, it is ready for publishing.
On this occasion I remember that the solution of the platform with “mixed configuration” known now as a “Spanish solution” (https://en.wikipedia.org/wiki/Spanish_solution) was operated in Warsaw (Poland) in 60., 70., and 80. of the 20. century (https://en.wikipedia.org/wiki/Warszawa_%C5%9Ar%C3%B3dmie%C5%9Bcie_railway_station).
Reviewer 2 Report
The authors addressed all my questions. It can be published in its current form.